# Photoplethysmography in Normal and Pathological Sleep

**DOI:** 10.3390/s21092928

**Published:** 2021-04-22

**Authors:** Ramona S. Vulcan, Stephanie André, Marie Bruyneel

**Affiliations:** 1Department of Pulmonary Medicine, CHU Brugmann, Université Libre de Bruxelles (ULB), 1020 Brussels, Belgium; ramona.vulcan@chu-brugmann.be (R.S.V.); stephanie.andre@chu-brugmann.be (S.A.); 2Department of Pulmonary Medicine, CHU Saint-Pierre, Université Libre de Bruxelles (ULB), 1000 Brussels, Belgium

**Keywords:** photoplethysmography, wearables, sleep-related breathing disorders, obstructive sleep apnea, polysomnography, heart rate variability

## Abstract

This article presents an overview of the advancements that have been made in the use of photoplethysmography (PPG) for unobtrusive sleep studies. PPG is included in the quickly evolving and very popular landscape of wearables but has specific interesting properties, particularly the ability to capture the modulation of the autonomic nervous system during sleep. Recent advances have been made in PPG signal acquisition and processing, including coupling it with accelerometry in order to construct hypnograms in normal and pathologic sleep and also to detect sleep-disordered breathing (SDB). The limitations of PPG (e.g., oxymetry signal failure, motion artefacts, signal processing) are reviewed as well as technical solutions to overcome these issues. The potential medical applications of PPG are numerous, including home-based detection of SDB (for triage purposes), and long-term monitoring of insomnia, circadian rhythm sleep disorders (to assess treatment effects), and treated SDB (to ensure disease control). New contact sensor combinations to improve future wearables seem promising, particularly tools that allow for the assessment of brain activity. In this way, in-ear EEG combined with PPG and actigraphy could be an interesting focus for future research.

## 1. Introduction to Photoplethysmography

Photoplethysmography (PPG) is an unobtrusive small device that is able to measure numerous physiological functions. Due to the fact that sleep follows an observed cardiovascular and respiratory pattern modulated by the autonomic nervous system, PPG is able to catch these changes and allow for the extrapolation of sleep staging and abnormal breathing pattern data. This data extraction process is complex, as analysis relies on mathematical models that should be tested, trained, and validated on a dataset before a potential clinical application [1]. The purpose of the present overview is to review the role of PPG in the detection of normal and pathologic sleep, to present the current limitations, to compare PPG with other wearables, and, finally, to assess its potential medical applications.

### 1.1. Historical Overview

The term photoplethysmography (PPG) is derived from ancient Greek. “Plethysmography” is derived from “plethysmos”, which means enlargement, and “graphein”, which means ‘’to write’’. In medicine, this term is commonly used for measuring and recording changes in the volume of the body, of an organ, or of a tissue. PPG is plethysmography that is obtained using an optical device (from “phôtós”, meaning “light”). Overall, the term describes an optically driven method used to detect blood volume changes (BVCs) in the microvascular bed of tissues [2].

The first attempts at developing instruments designed to monitor BVCs date from 1936 with the work of two groups of American researchers: Molitor and Kniazuk from the Merck Institute of Therapeutic Research in New Jersey, and Hanzlik et al. from Stanford University School of Medicine [2].

In 1937, another American, Alrick B. Hertzman from the Department of Physiology at St. Louis University, published a first description of a photoelectric plethysmograph that could transcutaneously measure peripheral BVCs in fingers and toes, during Valsalva, vasodilator, and vasoconstrictive maneuvers, and he became the first to introduce the term PPG [3].

In 1940, Hertzman and his research team were able to split the PPG waveform into two components with separate electronic amplifiers for the alternating current (AC) and direct current (DC) components. These pulsatile and non-pulsatile components of transmitted light enabled Japanese engineer Aoyagi to develop the first pulse oximeter 30 years later [4].

At the same time in London, Squire was the first to realize that the transmission of red and infrared light through tissue changed with oxygen saturation [4]. In 1949, Wood, at the Mayo Clinic, extended and mathematically developed the ideas of Squire to provide the basis for the emerging final work of Aoyagi who was able to compute arterial saturation by looking at the ratio between AC and DC at two different wavelengths, red and infrared (IR) [4]. Pulse oximeters have been available commercially since 1983 and are still the standard procedure for oxygen saturation estimation and heart rate (HR) measurement [3].

### 1.2. Technical Aspects of Photoplethysmography

The method relies on the observation that the light travelling through different biological tissues is absorbed in different proportions by its components, such as skin pigments, bone tissue, or red blood cells. Fluctuations in the absorbed light are determined by equivalent quantitative fluctuations of the light-absorbing tissues, mainly due to fluctuations in blood flow. These cardiac cycle-reflecting fluctuations were first described by Akbar Mohamed, who established the foundation of pulse wave analysis with his sphygmogram at the end of the nineteenth century and pointed out the difference between peripheric (radial) and central (carotid) waves [5].

These waves occur mainly in the arterial segment of the vascular bed and, to a lesser extent, in its venous segment, with an increase in blood volume during cardiac systole and a reciprocal decrease during cardiac diastole. The PPG sensors detect these BVCs, measured as variations in the intensity of the transmitted or reflected light from the microvascular bed of the investigated tissue. The PPG waveform includes two components: DC, the relatively steady component, that slowly varies with respiration and relates to the tissues and to the average blood volume in the venous blood and the steady arterial flow, and AC, the pulsatile physiological waveform [6] related to the cardiac cycle, that reports BVCs between the systolic and diastolic phases. The AC component frequency depends on the HR and is superimposed onto the DC component. Pulse oximeters use electronic filtering and amplification to separate the AC and DC components for estimating peripheral oxygen saturation (SpO2) and for extracting the PPG signal [2].

Figure 1 summarizes the acquisition of the PPG signal. The AC arterial systole-related BVCs component is superimposed onto the slowly varying DC, tissue, venous and arterial diastolic steady blood volume-related component. The variability of the DC component is related to respiration and vasomotor activity.

The interaction of light with biological tissue involves scattering, absorption, and/or reflection. The choice of the right wavelength is essential to obtaining a PPG signal. Red light (approximately 660 nm) and near-IR light (approximately 940 nm) pass easily through human tissue and have been routinely used as a light source in PPG sensors. PPG relies on the difference in measured absorption between oxygenated and deoxygenated hemoglobin at the two wavelengths [7].

More recently, because of their wide intensity fluctuations in modulation during the cardiac cycle, green-wavelength PPG devices have been developed and are increasingly used. The new green light-emitting diodes (LEDs) also have a considerably higher absorptive capacity when compared to IR light, for both oxyhemoglobin and deoxyhemoglobin. Currently, green light provides the strongest PPG signal. Two types of signals can be obtained from wearable PPG: transmissive absorption (as at the finger-tip) or reflective (as on the forehead or wrist). In transmissive PPG, the LED-emitted light is transmitted through the tissue and detected by the photodetector (PD) placed on the opposite side of the tissue, while in reflectance mode, the PD lies on the same side as the LED source, detecting the reflected light from the tissue (Figure 1).

In order to be effective, the PPG sensor must be placed on specific body spots where the anatomic constitution and arrangement of the tissues to be traversed assure a clear transmission from the LED to the PD. The preferred sites are the fingertip and the earlobe, but other body sites, such as the nasal septum, the cheek, or the tongue have also been used.

However, there are some limitations: these sites have limited blood perfusion, they can be exposed to low ambient temperatures, and interfere with daily activities. Reflectance mode eliminates the problems associated with sensor placement, and a variety of measurement sites can be used, but motion artifacts and pressure disturbances can also limit the measurement accuracy of physiological parameters. Other limitations influencing SpO2 signal quality include hypothermia, arterial vasoconstriction, low cardiac output (<2.4 L/min/m^2^), and elevated cutaneous vascular resistance [8].

### 1.3. Recent Developments in Photoplethysmography Technology

Significant advances in PPG probe design have occurred in the last few decades with progress in modern electronics and semiconductor technology leading to the extensive use of LEDs, photodiodes, and phototransistors. There have also been numerous developments in computer-based digital signal processing and pulse wave analysis. A big step forward was taken with the development of small, wearable, pulse rate (PR) sensors, leading Jonathan and Leahy to report HR estimation using smartphones in 2010 [9]. Ever since, numerous studies have been conducted by groups whose objective was to obtain robust physiological information using small wearable unobtrusive devices as described later in this paper (see Section 3).

The second foundational stone of the expansion of the PPG technique in the bioengineering field was laid in 2000 when Wu et al. [10] proposed the first system for noncontact imaging photoplethysmography (iPPG), a pioneering system that uses a camera to detect pulsatile changes in optical properties of the skin. Over the years, this method has been described in literature under different names such as remote PPG (rPPG), non-contact PPG (ncPPG), imaging PPG (iPPG), and PPG imaging (PPGi/PPGI) [11]. iPPG using digital cameras has generated increasing scientific interest with more than 60 papers published between 2010 and 2015.

Over the last few years, the iPPG technique has been increasingly tested in various settings and numerous papers are available on its performance in clinical applications. iPPG offers promising perspectives in providing a comfortable alternative to monitoring in infants, elderly people, patients with chronic pain, particularly neuropathic pain or migraine, with burnt skin, and patients undergoing dialysis. The technique has also been tested in peri-operative settings, in anesthesia monitoring, in critical patient monitoring, in arrythmia detection, and in burn care [12]. Van Gastel et al. recently described the possibility of obtaining contactless vital sign monitoring during sleep, opening the possibility of sleep apnea and sleep disorders assessment through rPPG [13].

In 2020, during the COVID pandemic, Casalino et al. proposed the use of a camera for non-contact and real-time measurement of blood oxygen saturation based on video face processing and rPPG with the enormous potential advantage to avoid physical contact in contagious patients [14].

## 2. Applications of Photoplethysmography in Clinical Physiological Measurements in Healthy Subjects

PPG is a multisignal provider: it provides not only SpO2, but also HR and respiratory rate (RR) values. The AC component of the PPG waveform is synchronous (but delayed with a transition time interval) with the systolic ECG complex, and, therefore, can be used as an HR surrogate. Signal extraction, clinical applications, and limitations are summarized in Table 1.

### PPG in Normal Sleep

Due to the variety of different signals that can be derived from PPG, researchers have studied the ability of PPG to describe sleep staging. Indeed, PPG allows for the recording of four physiological parameters (HR, SpO2, BP, RR) or surrogates, capturing physiological changes that are correlated with brain activity. During normal sleep, the autonomic nervous system (ANS) modulates cardiovascular functions during sleep onset and the transition to different sleep stages. The analysis of heart rate variability (HRV), which can be extracted from PPG, is a reliable tool to assess cardiovascular autonomic control as it can report physiological autonomic changes present during the wake-to-sleep transition, sleep onset, and different sleep stages: REM and NREM sleep [15,16].

The gold standard for the determination of sleep staging, visually represented by the hypnogram, is polysomnography (PSG). PSG allows for the scoring of the different stages of sleep including the waking, rapid eye movement (REM), and non-rapid eye movement (NREM) stages. The NREM stage is further divided into N1, N2, and N3. PSG is, unfortunately, an expensive, intrusive, and time-consuming in-hospital sleep lab procedure [17,18]. Moreover, access to sleep labs is limited, at least in some countries [19,20].

Sleep disorders are growing in the general population. Insomnia and obstructive sleep apnea (OSA) are the most common sleep disturbances. A significant proportion of OSA remains undiagnosed [21], emphasizing the need for early diagnosis and treatment. To overcome the problems of accessibility, complexity, and costs, many simplified portable monitoring devices (PM) have been developed since the 1980s to perform home sleep apnea testing (HSAT). In 1994, the American Academy of Sleep Medicine (AASM) classified, for the first time, all sleep recording devices. According to the number of sensors, the scientific society categorized them from level I to IV [22]. Type II refers to unattended PSG and is rarely used. HSAT refers to type III and IV.

The main difference between HSAT and PSG is the lack of electroencephalogram (EEG) recording associated with HSAT devices, which makes it impossible to score sleep stages and to distinguish sleep periods from wake periods. These recordings report time in bed or a recording period.

Wearables, belonging to the type IV category, have gained in popularity during the last few decades. These devices (including PPG) can be used to estimate sleep but are also used in a broader set of applications among the general population. Indeed, there is a large focus on the self-monitoring of physiological parameters in daily living, with the purpose of improving the self-management of sleep duration, sleep schedule, and healthier behavior for the primary prevention of cardiovascular and metabolic disorders induced by sleep reduction/sleep fragmentation [23]. In contrast to PSG, wearables allow for prolonged periods of evaluation. Currently, most smartphones and wearable health devices (e.g., wrist-worn watches) include accelerometers and provide information about sleep duration and sleep staging. Unfortunately, few have been validated against PSG and the performance of these devices for sleep stage detection is poor [24,25,26]. The characteristics, benefits, and limitations of commercialized wearable sleep-trackers (meaning the use of contact sensors) for sleep detection in adults are summarized in Table 2 [24,27,28,29,30]. Sleep stage detection performance is not reported in the table as there is considerable heterogeneity between studies and the results are generally unreliable.

Now let us review the recent academic studies focused on PPG signal-based algorithms to document sleep architecture in normal sleep. For this review, we have selected studies that compared data extracted from PPG with PSG in healthy adults, with the purpose of sleep/wake detection and/or classification of sleep stages.

In order to understand these new diagnostic methods in clinical practice, the statistical interpretation of studies should be well understood. Sensitivity is defined as the ratio of true positive samples to the total number of positive samples (true positive + false negative), while specificity is defined as the ratio of true negative samples to the total number of negative samples (true negative + false positive). Therefore, a sensitive tool means, in the present case, that you will not miss a sleep period or a specific sleep stage. A specific tool means that we can be confident in a negative result (specificity = true negative/negative samples). Negative predictive value (NPV) is the ratio of true negative and the sum of true negative and false negative, whereas positive predictive value (PPV) is the ratio of true positive and the sum of true positive and false positive. Accuracy is also often used and is defined as the proximity of measurement results to the true value. The accuracy of a measurement system is the degree of closeness of measurements of a quantity to that quantity’s true value. Studies often report Cohen’s kappa coefficient (κ) that represents the degree of accuracy and reliability in a statistical classification and measures the agreement between two raters (judges) who each classify items into mutually exclusive categories. Interpretation of Cohen’s kappa results is as follows: 0.01–0.20: slight agreement; 0.21–0.40: fair agreement; 0.41–0.60: moderate agreement; 0.61–0.80: substantial agreement; 0.81–1.00: almost perfect or perfect agreement [31]. Of note, when we compare manually scored PSG to other sleep measurements, AASM inter-scorer agreement between two sleep experts is about 0.78 [32]. Finally, the area under the curve (AUC)–receiver operating characteristics (ROC) curve is a performance measurement for the classification problems at various threshold settings.

Eyal et al. [33] aimed to validate an automated sleep analysis that was based on the inter-beat-interval (IBI) series obtained from transmissive PPG, and that used features of HRV. The algorithm was tested against the gold standard, PSG. PPG allows for the detection of instantaneous IBI. The research team developed sleep diagnostic software that relied on IBI series obtained from an electrocardiogram (ECG) signal (IBIECG). With the wide availability of PPG for HR detection, their interest was in evaluating whether the IBI obtained from PPG signals (IBIPPG) during sleep allowed for the evaluation of sleep structure. Indeed, previous studies [34,35] indicated that heart rate variability (HRV) based on IBIPPG could be used as an alternative to the HRV calculated from electrical signals of the heart, IBIECG. Data from 88 PSGs that included both ECG and finger PPG were used to check the performance of a PPG-based algorithm. They found that epoch-by-epoch sleep/wake comparisons between the output of the IBIPPG automated algorithm with PSG resulted in *fair agreement*, with sensitivity to wake stage (W) as low as 38%, but a very high specificity: 92%. When considering sleep stages only, the capacity of the IBIPPG developed algorithm to distinguish between REM stages and NREM stages showed a sensitivity to REM sleep as low as 51% and a specificity of 92%, with κ = 0.46. The sensitivity between different sleep stages in epoch-by-epoch scoring using the developed algorithm vs. gold standard were for wake/sleep; REM/NREM deep sleep; REM/NREM light sleep 38%, 51%, 77%, and specificity 92%, 92%, 75%, κ = 0.31, 0.46, 0.42, respectively. This algorithm was an interesting first attempt to establish sleep architecture based on PPG signals, but accuracy was low and suggested that further improvement was needed.

Motin et al. [36] designed an automated approach, extracted from time domain features, for sleep–wake classification based on fingertip PPG signals (SpO2, surrogate cardiac signals). A support vector machine was used on a training dataset to teach the machine to distinguish sleep and wake stages, and this was then applied to 2818 sleep–wake events from PSG. The performance was interesting, with a sensitivity of 81%, specificity of 82.5%, and accuracy of 81%, opening the door to further use of this model.

Another study added actigraphy data to PPG signals in order to improve sleep/wake and sleep staging detection accuracy. Fonseca et al. [27] added accelerometer measurements to PPG signals (PPG-derived HRV) to assess sleep. They obtained, in 51 healthy participants, an agreement of 70% for total sleep time (TST), total wake time, sleep efficiency, and wake after sleep onset (WASO). Performance was lower in individuals with long sleep onset latency (SOL). Better results were obtained for distinguishing sleep from wake stage (accuracy 91%, κ = 0.55) than for classifying wake/NREM/REM (accuracy 73%, κ = 0.46) or wake/N1 + N2/N3/REM (accuracy 59%, κ = 0.42). In contrast to previous studies, in this study, a wrist-worn reflective PPG was used. Potential sources of artefacts in this study are thus different and could be related to wrist movements or skin tissue motions.

Walch et al. [28] studied the use of a consumer-based wearable, the Apple Watch, and collected raw acceleration data and HR with their own mobile application. They also explored the benefit of mathematical models exploring the circadian clock to sleep/wake classification algorithm. The purpose was to test different algorithms for sleep staging and to compare the results with PSG. Three characteristics were used as raw data for the tested classification algorithms: motion (data obtained from the raw acceleration), HR, and a computed circadian estimate. The first step was to train their neural net model applied to an Apple Watch on a dataset from 31 healthy subjects. Compared to PSG, their mathematical model demonstrated a sleep/wake distinction with a sensitivity of 93% for the sleep (S) epochs scored and of 60% for the W epochs and a REM-NREM sleep stage differentiation accuracy of 72%. They then applied these algorithms to a dataset from the multi-ethnic study of atherosclerosis (MESA) cohort (188 subjects), a database that contains motion and HR information obtained from actigraphy, oximetry and co-recorded PSG. Their neural net sleep/wake classifier, trained using all features on the entirety of the Apple Watch dataset and tested on the MESA subcohort, showed a sensitivity of 60% for the W epochs, and 90% for the sleep epochs, and a κ of 0.525. The wake/NREM/REM neural net classifier achieved the best accuracy of 69%, and a corresponding κ = 0.4. This study highlights the possibility of using data extracted from consumer wearables and combining it in different settings to obtain better accuracy.

Beattie et al. [29] reported on the accuracy of an automated algorithm aimed to identify sleep stages starting from a wearable wrist-worn device equipped with a 3D accelerometer (Fitbit Surge, Fitbit) and with an optical pulse PPG (reflective PPG, green LED), compared to simultaneously recorded unattended PSG, in 60 adult participants. To train the classifier and then validate it, features such as movement, breath, and HRV were extracted from the accelerometer and PPG sensors. The overall per-epoch accuracy of the automated algorithm was 69%, with a κ = 0.52. Agreement between PSG and PPG for deep sleep, light-sleep, and REM were 62%, 69%, and 72%, respectively. The most common misclassifications were light/REM and light/wake mislabeling. The authors stressed the significant advantage of adding optical pulse signals to accelerometers to improve accuracy.

Finally, Zhao et al. [37] tested a reflective PPG-based multi-class automatic sleep staging (PMSS) method against PSG in a mixed population of subjects and patients. PSG signals from more than 27,000 periods of 27 subjects were used to extract PPG signals as data. The population was mixed, including 4 healthy subjects, 8 patients with REM sleep behavior disorder, 10 patients with nocturnal frontal lobe epilepsy, and 5 patients with insomnia. PPG data were initially preprocessed and the domains—time, frequency and nonlinear—were used for the extraction of a total of 21 features that were subsequently used by a Light Gradient Boosting Machine (Light GBM) classifier for multi-class sleep staging. Sleep staging classification was tested in different settings: 3-class (wake, NREM, REM), 4-class (wake, light sleep, slow wave sleep, REM), 5-class (wake, N1, N2, N3, REM), showing accuracy rates of 86% (κ = 0.79), 77%, and 72% (κ = 0.6), respectively. The sleep staging ability was the best for healthy people, with an accuracy rate of more than 80%. It remained, however, acceptable in patients, in whom the consistency was more than 0.60. Some errors were related to reflective PPG: when this method is used at a high light intensity, large errors can occur. The authors concluded that the model is still suitable for sleep staging of patients with sleep disorders.

Altogether, data related to sleep assessment by PPG alone or combined with accelerometer remain scarce and moderately accurate, especially for five-state sleep staging. Several limitations have been highlighted regarding signal capture and/or processing. Generalization of usage seems currently very limited by the heterogeneity of devices and signal processing.

Limitations also emerged from these different studies. These were related to the peripheral aspects of PPG signal analysis, whether we are talking about the transmission of the information to the peripheral detector, the signal detection and capture, or the interpretation. Thus, bias could be observed in settings of arterial vasoconstriction, hypothermia, motion artefacts [38], oxymetry signal failure (e.g., CO-hemoglobin > 3%, skin pigmentation, nail polish, advanced age, severe onychomykosis, dirty fingers, Raynaud’s disease, cold fingers, methemoglobin > 0.5%, decreased capillary perfusion), or a biased signal interpretation algorithm.

## 3. Photoplethysmography Applications in Sleep-Disordered Breathing Diagnosis

### 3.1. In Obstructive Sleep Apnea

Obstructive sleep apnea (OSA) occurs in the presence of repeated episodes of upper airway collapse during sleep. These events induce increasing respiratory efforts during the obstructive event. As upper airway resistance increases, either the rib cage or the abdominal wall begins to move out of phase. Pleural pressure decreases in parallel with inspiratory efforts and can lead to reduced left ventricular stroke volume [39]. Pleural pressure swings are believed to be the central mechanism generating arousals in OSA. OSA induces intermittent hypoxia/hypercapnia and increased activation of the sympathetic nervous system, leading to decreased long-term HRV [40]. PPG can thus be used to detect OSA. Obstructive respiratory events are shortly followed by a reactive relative bradycardia, an increase in PPG pulse amplitude and HR, as well as a post-obstructive severe vasoconstriction. During a hypopnea (only a partial upper airway obstruction), the force of respiratory muscles is increased and respiratory-induced intensity variation (RIIV) is more prominent. Pulse transit time (PTT), defined as the time needed for the pulse pressure wave to travel from the aortic valve to the periphery, and measured as the time delay between the R-wave on the ECG and the foot of the anacrotic wave recorded on the peripheral PPG waveform, is inversely proportional to BP. In case of BP decrease, PTT increases. A close relationship between the increase in esophageal pressure due to episodes of upper airway resistance (reflecting pleural pressure thus inspiratory efforts) and a progressive rise in the amplitude of PTT oscillations between inspiration and expiration has been well established in OSA [41,42].

The following section aims to review the studies that have compared data extracted from PPG with PSG in OSA. Of note, we have intentionally excluded from our analysis the studies that compared single oximetry to PSG for a lack of sensitivity and specificity for the detection of mild (apnea-hypopnea index (AHI) 5–14), moderate (AHI 15–29), or severe OSA (AHI > 30) even in high-risk populations [43]. For example, one study compared oximetry against unattended PSG. Using a cut-off of AHI ≥ 15 and AHI ≥ 30, oximetry had an accuracy of 86% and 74% in a high-risk population, but fell in a low-risk population, where accuracy reached 80% and 63%, respectively [44]. This was even lower in other studies.

In addition, new approaches for analyzing OSA-related patterns in high temporal resolution pulse oximetry have been developed during the last few decades and aimed to improve classification and outcomes of OSA. They have been recently extensively reviewed [45]. The purpose here is to focus on studies where SpO2 signal was tested in conjunction with data extracted from PPG.

### 3.2. Obstructive Sleep Apnea and Hypopnea Detection

Several studies have focused on mathematical algorithms to extract different parameters to detect obstructive sleep apnea and hypopnea. For example, Deviaene et al. [46] have combined a point-process model of heartbeat interval dynamics (to estimate ANS activity) with SpO2-based apnea detection to measure AHI, in comparison with PSG, in 102 patients suspected of having OSA. Based on six PPG features, they were able to demonstrate an accuracy of 83% to detect obstructive/central apnea and hypopnea (sensitivity of 74%, specificity of 87%). Compared to PPG or SpO2 alone, the performance of the combination of signals was better, except for SpO2 sensitivity (75%).

Barak-Shinar et al. [47] studied a pulse oximeter system (Morpheus Ox) with an automated analysis based on the PPG signal for the diagnosis of sleep-disordered breathing (SDB) in 140 sleep laboratory patients who were referred for sleep-disordered breathing. Each patient underwent an overnight PSG. An automatic analysis based on PPG and SpO2 signals allowed an AHI calculation. The gold standard test recorded a wide distribution of AHI values, ranging from 0 to 97 with a median value of 10.5. The sensitivity and specificity of PPG for AHI ≥ 5 and ≥ 15 levels were 94.4% and 96.5%, respectively. There were only six mismatching records out of the 140 total, concerning the AHI comparison, three false positive and three false negative. The detection of respiratory events (apnea or hypopnea associated with a saturation reduction of 4%) showed a sensitivity of 86.9% and a positive predictive value of 84%. Sleep and wake evaluation (epoch-by-epoch classification) showed an agreement of 77.88%. The limits of this study included its sleep lab setting and unknown reproducibility at home, possible oxymetry signal failure, and the inability of the system to detect hypopnea associated with arousal and not with oxygen desaturation.

Li et al. [48] aimed to test the accuracy in OSA diagnosis of an automated analysis of the PPG signal recorded by a standard pulse oximeter during the PSG, in patients with suspected OSA, as compared to PSG. Forty-nine OSA patients were included in the study. The PPG-derived results were compared with PSG-derived results for agreement tests. They found a significant correlation between the respiratory events index (REI) derived from PPG with the PSG-derived AHI (r = 0.935, *p* < 0.001), as well as between the PPG-oxygen desaturation index (ODI) and the PSG-ODI (r = 0.933, *p* < 0.001). The algorithm performed as follows: for mild OSA, PPG obtained a sensitivity of 95% but a specificity as low as 50%. For moderate OSA, the specificity was slightly lower (90%) but the specificity was slightly higher (65%). Their automated analysis performed best in severe forms of OSA, with a sensitivity of 90% and a specificity of 97%. Further, the ODI derived from PPG-based monitoring (PPG-derived ODI) highly correlated with its equivalent PSG-derived ODI (r = 0.933, *p* < 0.001) and the TST derived from PPG-based monitoring (PPG-derived TST) with the PSG-derived TST (r = 0.418, *p* = 0.003), but PPG-derived TST was underestimated (324 vs. 381 min, *p* < 0.001). The authors concluded that improvement of PPG-derived TST estimation is needed.

More recently, Papini et al. [49] tested a deep learning model using raw cardiorespiratory information (HRV and surrogates of respiratory activity) and sleep information from an rPPG signal obtained from a wrist-worn device. Their algorithm was trained on 252 recordings and aimed to best estimate AHI values. After completing the training period, they tested this algorithm on 188 clinical PSG recordings (clinical population, comprising healthy subjects and patients with various types and levels of disordered sleep). AHI from rPPG and PSG showed good agreement with a correlation of 0.61, and an error of 3 ± 10 events/h, so the model showed good performance in OSA screening (ROC–AUC = 0.84/0.86/0.85 for mild/moderate/severe OSA) as well as in assessing OSA severity (κ = 0.51). However, considerable underestimation of AHI occurred in 30 participants, but OSA classification was only slightly affected. Forty-two percent of the false-positive detections were because of epochs with artefacts such as limb movements that were probably misinterpreted as respiratory events. The advantage of this method to estimate AHI using rPPG is the possibility of implementing it on devices such as smartwatches and fitness trackers. According to the authors, the algorithm can be used for OSA screening and OSA severity estimation, even in a population with heterogeneous sleep disorders and cardiovascular comorbidities. They also stressed that the device gives similar results to HSAT for OSA severity estimation but with a much simpler device.

Hayano et al. [50] also explored a pulse rate-based algorithm, extracted from a PPG wearable watch device to detect OSA, in 41 patients undergoing diagnostic PSG for OSA. An automated algorithm adapted from an algorithm developed for ECG analysis to detect a characteristic HR pattern related to sleep apnea episodes, called auto-correlated wave detection with adaptive threshold (ACAT), was applied on PPG pulse interval data obtained from 41 patients undergoing diagnostic PSG. In the study group, the median AHI was 17.2 and 54% subjects had AHI ≥ 15. The hourly frequency of CVHR (Fcv) detected by the ACAT algorithm was closely related to AHI value (r = 0.81). The Fcv was greater in subjects with AHI ≥ 15 (19.6 ± 12.3/h) than in those with AHI < 15 (6.4 ± 4.6/h) and could distinguish between the two categories then with 82% sensitivity, 89% specificity, 81% NPV, 90% PPV, and 85% accuracy. The algorithm had the same performance as that applied to ECG R-R intervals during PSG. For the first time, this study demonstrated the equivalence of PPG and ECG for the detection of OSA. This seems to be due to the characteristics of CVHR and to the features of the ACAT algorithm. Here, also, the authors emphasized the great potential for using this social resource as a cost-effective large-scale screening for OSA. The main limitations were that the pulse wave signals were analyzed by the manufacturer’s cloud application via the Internet, which often remains out of the control of the physician conducting the test.

### 3.3. Sleep Staging in OSA Patients

Another research aim for the development of PPG is to test the ability of PPG for sleep staging detection in disordered sleep. Recently, Korkalainen et al. [51] used the PPG signal from a basic pulse oximeter (Nonin Xpod 3011) as a basis to develop and train an automated deep learning model aimed to estimate sleep staging in 894 suspected OSA patients. The algorithm, a combined convolutional and recurrent neural network, was trained individually for three-stage (wake/NREM/REM), four-stage (wake/N1 + N2/N3/REM), and five-stage (wake/N1/N2/N3/REM) classification of sleep. The tested models achieved an epoch-by-epoch satisfying accuracy, as follows: the three-stage model had 80.1% accuracy (κ = 0.65), the four-stage model had 68.5% accuracy (κ = 0.54), and the five-stage model had 64.1% accuracy (κ = 0.51). The five-stage model underestimated the TST with a fair mean error of 7.5 min. The mean AHI in the studied cohort as calculated from the PSG was 24.2 events/h while the simulated polygraphic AHI was 18.8 events/h. The three-stage PPG-based model calculated a mean AHI of 23.3 events/h, the four-stage model obtained 23.1 events/h, and the five-stage model obtained 22.6 events/h. The mean difference between the AHI calculated from PSG and from polygraphy was −5.3 events/h (*p* < 0.001), while the mean difference between the AHI calculated from PSG and PPG was −0.9 to −1.6 to events/h with the three-to-five stage model. Finally, good accuracy was obtained for sleep staging with the three-stage model and excellent agreement for AHI detection in this large OSA series. The authors stressed the possibility of applying PPG-based sleep staging during HSAT, as PPG is included in such PM, in order to increase the accuracy and clinical assessment of OSA (e.g., REM-related OSA).

### 3.4. In Central Sleep Apnea

One study has assessed PPG for Cheyne–Stokes breathing detection [52]. The researchers compared event detection and classification obtained by PSG (ambulatory and in hospital) and by an automated new algorithm system extracted from PPG and oxygen saturation (Morpheus Ox) in 74 subjects. The developed algorithm scored very well in detecting and scoring SDB and Cheyne–Stokes breathing (CSB) as compared to PSG results. Fifty percent of patients exhibited CSB. The sensitivity of CSB detection was 92% and specificity was 94%. AHI values ranged from 0 to 88 with a median of 29. Comparison of AHI ≥ 15 showed a sensitivity of 98% and specificity of 96%. Respiratory events were detected with an agreement of approximately 80% between the two methods, as calculated by regression and Bland–Altman plots. The developed method allowed for an acceptable analysis of sleep-and-cardiac-related breathing disorders. The sleep and wake comparison between the system and the gold standard yielded 75% agreement. These encouraging results support the notion that this system could be considered a suitable tool for SDB screening in patients with cardiovascular disease.

## 4. Discussion

Physiological signals captured by PPG are useful for estimating sleep duration and sleep structure, even in patients suffering from OSA. However, when compared with PSG, the discriminative power of PPG to accurately capture sleep architecture remains limited. The addition of accelerometry increases sleep staging detection, but this combination of sensing modalities is unable to identify all stages of sleep and its accuracy is also lower compared to sleep EEG. Detection of AHI in OSA is also feasible with moderate agreement with PSG.

The fundamental limitation of PPG for use as an accurate sleep monitoring method is its lack of ability to capture EEG activity. Other limitations include aspects of signal acquisition (e.g., oxymetry signal failure, motion artefacts) and signal processing. Signal acquisition is likely to improve in the future by integration of new types of sensors to better identify motion. Indeed, researchers have developed the fluidic fabric muscle sheet, a miniaturized device that allows for the very sensitive measurement of motion [53]. This could certainly help in fine-tuning motion capture, as a complementary sensor to actigraphy. Innovative oxymetry sensors have also been developed. The reflectance oximeter array is a flexible 2D printed electronic system that is able to overcome PPG signal capture limitations related to shock or low blood perfusion. With this device, Khan et al. were able to determine oxygenation in the absence of pulsatile signal [54].

Regarding signal processing, machine-learning methods are very useful in disease prediction but also show bias related to possible missing values in datasets or to the exclusion of outliers [1]. Generalization of usage of mathematic models seems currently limited by the heterogeneity of devices and signal processing, but progress can be made by sharing algorithms between researchers and large-scale population-based studies to improve sleep stage detection in normal and pathologic sleep.

The main benefit of the PPG technique in medical applications is likely to be for the long-term monitoring of the sleep-disturbed population (e.g., insomnia patients or in cases of circadian rhythm sleep disorders), particularly for the assessment of changes in sleep patterns after treatment. The advantage of using PPG combined with accelerometry rather than accelerometry alone is to be able to obtain better sleep stage classification, but this needs to be confirmed in future large-scale studies in patients.

Regarding SDB, the main purpose of these tools is to detect OSA in home settings, for triage, or for treatment trials, in high-risk populations [55]. Cheyne–Stokes breathing detection could also be very useful in selected populations (e.g., heart failure), as patient management is very different from OSA [56]. It would also be very useful to obtain long-term sleep monitoring in treated SDB, with the aim of assessing disease control, and in OSA, but also in more severe populations where hypoxemia resolution is more challenging (e.g., overlap syndrome (OSA associated with chronic obstructive pulmonary disease) or obesity-hypoventilation syndrome provoked by OSA) [57,58].

In the field of OSA diagnosis, the major pitfall of using HSATs is the occurrence of false-negative results, creating the need for test repetition, which should be PSG in this case [59]. PPG is no exception to this rule. Even if there are prospects for development and potential medical applications, the literature surrounding PPG is currently very limited and needs to be further developed. To increase the accuracy of PPG for sleep stage assessment, one proposal for development could be to add a single EEG channel to better capture brain activity in normal and pathologic sleep. This would also allow for the detection of arousals. For example, an in-ear EEG sensor has been recently explored. The largest series, published by Nakamura et al., explored in-ear EEG in 22 patients [60]. Agreement for sleep stage classification was 74% (κ = 0.61). Other small series have also reported interesting results, with accuracies between 72% and 90% [61]. Recently, technical improvements of in-ear EEG were obtained by Kaveh et al. with the development of the first wireless, dry multielectrode in-ear EEG [62].

The ear canal is a very interesting location for recording physiological parameters during sleep. Indeed, reflective PPG can also be obtained accurately from this location and it avoids the difficulties of capturing SpO2 at the finger, ear lobe, or wrist in case of motion or compromised peripheral perfusion [63,64]. Future research should investigate in-ear combined EEG and PPG in normal and pathologic sleep. This will require technological advances to obtain ultra-low-power processors and integrated miniaturized electronics for both sensors.

Another interesting location to capture both EEG and PPG is the forehead. Reflective PPG can be easily obtained [65] and could be coupled with one frontal EEG lead. Some studies have explored the added value of one frontal EEG to HSAT, analyzed by automated scoring algorithms. Forehead EEG increases OSA detection by providing reliable wake/sleep identification, leading to better AHI calculation [66,67,68].

Other combinations of contact sensors could also potentially increase PPG accuracy. For example, combining skin temperature measurements with PPG could also help to better differentiate sleep and wake, as the onset of sleep is preceded by a rise in skin temperature [30]. This aspect should be further studied. The addition of a microphone to PPG could be also interesting. Indeed, snoring sounds have been shown to provide a good estimation of sleep architecture and quality [69]. In addition, machine learning is offering additional opportunities for specific uses of snoring characteristics, such as snoring frequency, which has been shown to relate to sleep apnoea [70].

To advance research, other issues need to be addressed. These wearable technologies require a substantial collaboration of medicine with engineering to assess medical and technical constraints arising from hardware integration and signal acquisition, as well as for various levels of signal processing and data communication. University and research teams need to collaborate to test and implement wearables on a larger scale, allowing for big data analysis, far from mysterious industrial settings, where algorithms often remain secret and are never shared with researchers. Consumers should be aware of the reliability of the device they use, “medical” (e.g., validated through academic research protocols) or “industrial” (usually not validated). This should also help avoid duplication in algorithm development and subsequent incompatibility of devices [71].

In conclusion, this review has outlined the role of PPG for unobtrusive sleep studies. PPG has specific interesting properties, particularly the ability to capture the modulation of the autonomic nervous system during sleep. Recent advances have been made in PPG signal acquisition and processing, including coupling PPG with accelerometry in order to allow the construction of hypnograms in normal and pathologic sleep and to detect sleep-disordered breathing (SDB), but the accuracy of these techniques remains limited. Several prospective developments to overcome PPG limitations (e.g., oxymetry signal failure, motion artefacts, signal processing) are under investigation. New sensor combinations to improve future wearables are also promising (e.g., EEG, skin temperature, microphone). Collaboration with engineers is mandatory for improving signal processing aspects on a large scale.

There is a wide range of potential medical applications for PPG, including home-based detection of SDB (for triage purposes), and long-term monitoring of insomnia, circadian rhythm sleep disorders (to assess treatment effects), and treated SDB (to ensure disease control).

## Figures and Tables

**Figure 1 sensors-21-02928-f001:**
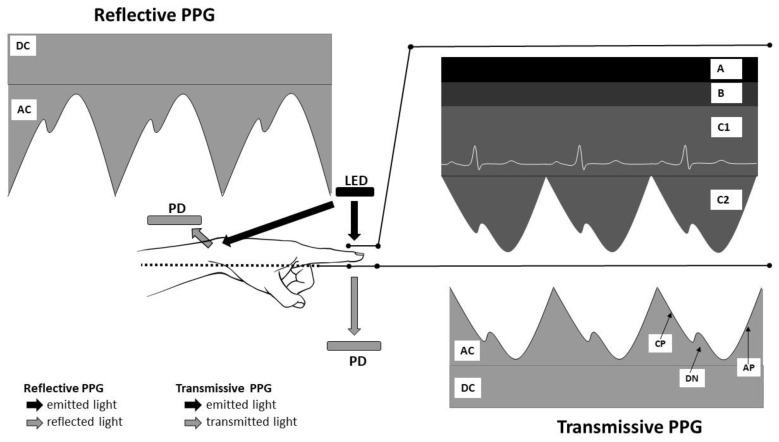
Basis of photoplethysmography (PPG) signal acquisition. LED: light-emitting diodes; PD: photodetector; A: other tissues; B: venous blood; C1: non-pulsatile component of artery blood; C2: pulsatile component of artery blood; AC: alternating current-the pulsatile part of the PPG waveform; DC: direct current—the steady part of the PPG waveform; AP: anacrotic phase; CP: catacrotic phase; DN: dicrotic notch.

**Table 1 sensors-21-02928-t001:** Applications of photoplethysmography in clinical physiological measurements in healthy subjects.

	Signal (Type and Processing)	Usefulness	Limitations
SpO2	AC and DC components of PPG	Ambulatory monitoring Hospital monitoring Anesthesia ICU	Dyshemoglobinemias Reduced accuracy for low values
Heart Rate	AC component of PPG Upsampling and algorithm to reject artefacts PPG-derived HR = pulse rate (PR)	Ambulatory monitoring Hospital monitoring Anesthesia ICU Neonatal care Sleep (HR variability)	Cardiac arrhythmias Movement artefacts
Blood pressure	DC component of PPG Surrogate pulse measure of BP = pulse transit time (PTT), calculated from ECG R wave to the foot of PPG pulse	Ambulatory monitoring Hospital monitoring Anesthesia ICU	Cardiac pre-ejection period to PTT
Respiratory rate	DC component of PPG Extraction algorithm to isolate respiratory induced intensity variation	Ambulatory monitoring Hospital monitoring Anesthesia ICU Neonatal care Sleep	

AC: alternating current, DC: direct current, PPG: Photoplethysmography, ICU: intensive care unit, HR: heart rate, SpO2: peripheral oxygen saturation, BP: blood pressure, ECG: electrocardiogram, PTT: pulse transit time.

**Table 2 sensors-21-02928-t002:** Wearable sleep-trackers dedicated to sleep measurement in healthy subjects.

Wearable	Technique	Overall Performance vs. Polysomnography	Limitations
actigraphy-Actiwatch 64, actiwatch spectrum 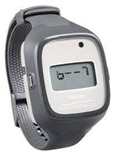 -Sensewear pro 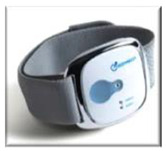	Accelerometer ± skin temperature	Overestimation of SE, TST, and underestimation of SOL, WASO	-High cost-Time-consuming interpretation by trained staff (missing data, motion artefacts to remove)
consumer wearables (first generation)-Fitbit, Fitbit ultra, Fitbit Flex 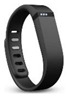 -Misfit Shine-Jawbone Up 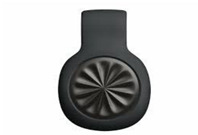	Accelerometer	Overestimation of SE, TST, and underestimation of WASO	-Commercial devices not developed for clinical use-Raw data and manufacturer algorithms are not accessible-Motion artefacts
consumer wearables (with sleep mode activation)-Jawbone Up Move-Withings pulse O2 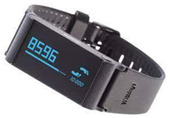	Accelerometer	Overestimation of SE, TST, and TIB	-Commercial devices not developed for clinical use-Raw data and manufacturer algorithms are not accessible-Motion artefacts
consumer wearables (last generation)-Fitbit surge-Fitbit charge 2-Apple watch 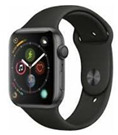 Devices under development (requiring validation vs. PSG):Corsano CardiowatchScanWatch Withings	Accelerometer skin temperature± PPG	Overestimation of SE, TST, and underestimation of SOL	-Commercial devices not developed for clinical use-Raw data and manufacturer algorithms rarely accessible-Motion artefacts-Failure of PPG signal capture

TST: total sleep time; TIB: time in bed; SE: sleep efficiency; SOL: sleep onset latency; WASO: wake after sleep onset; PPG: Photoplethysmography; PSG: polysomnography.

## Data Availability

Raw data are available for presentation to the referees and the editors of the journal, if requested.

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
