# Peer review of "Photoplethysmography in Normal and Pathological Sleep"

_sensors, 2021, doi:10.3390/s21092928_

Round 1
Reviewer 1 Report
Photoplethysmography in normal and pathological sleep
Ramona S. Vulcan et al.
Photoplethysmography (PPG) is one of the most commonly used techniques for noninvasive measurements of physiological process. It is also used in determining staging in sleep among other techniques. In this article, Vulcan et al. presents an overview of the PPG devices and its use in unobtrusive sleep studies. The authors presented a historical overview of the PPG methods, their developments, and the general mechanisms of the PPG technique. Finally, they discussed a few prospects of existing technologies and indicated potential future research directions.
Comments to the authors:
The scientific presentation of the article is below average to poor. The authors often reviewed the earlier research in a descriptive manner, only discussing what others have done without explaining the underlying scientific contribution or novelty of that article and why that is important. This approach reduces the impact of the article and often becomes boring to read.
Furthermore, I felt that the article does not have a story. The authors collected some prior research and presented those without building a proper story. The article doesn’t read well.
A major drawback of this article is that the authors often did not cite the refer articles properly. Many of the texts refer to some prior literature without a citation. Here are a few examples…
At the same time in London, Squire was the first to realize that the transmission of red and infrared light through tissue changed with oxygen saturation. In 1949, Wood, at the Mayo Clinic, extended and mathematically developed the ideas of Squire to provide the basis for the emerging final work of Aoyagi who was able to compute arterial saturation by looking at the ratio between AC and DC at two different wavelengths, red and infrared (IR).[ref] Pulse oximeters have been available commercially since 1983 and are still the standard procedure for oxygen saturation estimation and heart rate (HR) measurement.[ref]
The first attempts at developing instruments designed to monitor BVC date from 1936 with the work of two groups of American researchers :: Molitor and Kniazuk from Merck Institute of Therapeutic Research inNew Jersey, and Hanzlik et al. from Stanford University School of Medicine.[ref]
In 1937, another American, Alrick B. Hertzman from the Department of Physiology of St. Louis University, published a first description of a photoelectric plethysmograph that could transcutaneously measure peripheral BVC in fingers and toes, during Valsalva, vasodilator and vasoconstrictive maneuvers and he became the first to introduce the term of PPG.[ref]
There are many similar examples.
Furthermore, the authors often randomly changes the discussion in a paragraph. This make the article difficult to read. Also it is not consistent. Here is an example among many…
A big step forward was taken with the development of small, wearable, pulse rate (PR) sensors, leading Jonathan and Leahy to report HR estimation using smartphones in 2010 [2]. Further developments occurred in 2000, when the first system for noncontact imaging photoplethysmography (iPPG) was proposed by Wu et al. [3]. Currently, there is rapid growth in the literature focused on the development of PPG techniques.
Research in 2000 cannot be as a results of “further development” of 2010! Also, the last sentence is very strange!
The authors presented an elaborative discussion on the early history of PPG and its’ developments. Some of these are interesting. However, I believe the article is missing a discussion about the general importance of the PPG technique in medical applications.
Also, authors should briefly mention what other techniques are available to solve the addressed issue. And, why PPG is a successful technique over others, if so.
Typical review article should present most recent progress in the field. This article largely oversights that. Specially in the wearable section. In recent years, this field has grown enormously and the authors rarely discussed any related cutting edge research. Prof. Arias’ group, Visells’ group, Rogers’ group and many others have remarkably contributed in this field. I believe, the authors should discuss this emerging area more elaboratively.
The authors added two tables in the article. However, surprisingly there is not enough images. It is often useful for the readers to visually inspect the devices. My suggestion would be to add more figures of cutting edge devices to improve the quality of the article.
To add another comment here. I believe that the three figures in the present manuscript can be combined in a single figure. That would help the reader to compare different techniques and the operational mechanisms of different approaches.
What are the fundamental limitations of PPG devices to use for sleep monitoring?
A table comparing different techniques for sleep monitoring and their benefits and limitations would be helpful.
Author Response
Dear reviewer,
Thank you for your excellent comments that have helped us to improve the paper. We have carefully adapted and reworked the manuscript accordingly.
M Bruyneel
Comments to the authors:
The scientific presentation of the article is below average to poor. The authors often reviewed the earlier research in a descriptive manner, only discussing what others have done without explaining the underlying scientific contribution or novelty of that article and why that is important. This approach reduces the impact of the article and often becomes boring to read.
-> We have tried to better highlight the contribution and novelty of previous work in the abstract and discussion
Furthermore, I felt that the article does not have a story. The authors collected some prior research and presented those without building a proper story. The article doesn’t read well.
-> We have reworked the discussion to present the different perspectives and medical applications related to the technique
A major drawback of this article is that the authors often did not cite the refer articles properly. Many of the texts refer to some prior literature without a citation. Here are a few examples…
-> It is true that a lot of the historical references are missing. Thank you for making us aware of this issue. We have added them to the manuscript.
Furthermore, the authors often randomly changes the discussion in a paragraph. This make the article difficult to read. Also it is not consistent. Here is an example among many…
-> We have added a new chapter to the introduction to better emphasize recent advances in PPG.
1.3. Recent developments in Photoplethysmography technology
à Research in 2000 cannot be as a results of “further development” of 2010! Also, the last sentence is very strange!
-> We have removed this faulty sentence.
The authors presented an elaborative discussion on the early history of PPG and its’ developments. Some of these are interesting. However, I believe the article is missing a discussion about the general importance of the PPG technique in medical applications.
-> We have reworked the discussion to focus on medical applications.
Also, authors should briefly mention what other techniques are available to solve the addressed issue. And, why PPG is a successful technique over others, if so.
-> We have presented wearables in the "2.1. PPG in normal sleep" chapter including a new table, and other techniques are also discussed further in the Discussion section.
Typical review article should present most recent progress in the field. This article largely oversights that. Specially in the wearable section. In recent years, this field has grown enormously and the authors rarely discussed any related cutting edge research. Prof. Arias’ group, Visells’ group, Rogers’ group and many others have remarkably contributed in this field. I believe, the authors should discuss this emerging area more elaboratively.
-> Thank you, we agree that these very interesting works were missing and these are now presented in the Discussion section.
The authors added two tables in the article. However, surprisingly there is not enough images. It is often useful for the readers to visually inspect the devices. My suggestion would be to add more figures of cutting edge devices to improve the quality of the article.
-> We have presented commercialized wearables in the "2.1. PPG in normal sleep" chapter in a new table, including numerous images.
To add another comment here. I believe that the three figures in the present manuscript can be combined in a single figure. That would help the reader to compare different techniques and the operational mechanisms of different approaches.
-> According to your suggestion, we have merged the figures.
What are the fundamental limitations of PPG devices to use for sleep monitoring?
-> The fundamental limitations are now explained in the Discussion section.
A table comparing different techniques for sleep monitoring and their benefits and limitations would be helpful.
-> We have developed benefits and limitations of wearables in the "2.1. PPG in normal sleep" chapter in a new table.
Reviewer 2 Report
Authors propose a review article on PGG applications for sleep analysis.
The article is well written. However, in my opinion the first part of the article, that is devoted to explain the PPG technique, is too much specific, and not suitable for a general reader. Please try to be more clear. Moreover, the introduction describe the history of PPG, but references are missing. Please add references for all the works that you have mentioned (e.g. The first attempts at developing instruments). Moreover, at the end of paragraph 1.1 you mentioned the use of new techniques for PPG acquisition. Please add recent references (e.g. https://doi.org/10.1109/ISCC50000.2020.9219718)
Figures 1,2 and 3 are not clear. Please use good quality images. Furthermore, if these images have been already published in other works, please add a reference to them
The meaning of table 2 is not clear. Please add a description of this table content in the text.
Please describe the process that you have followed to select the articles to include in this review.
Please proofread the article for typos and use the same font style (e.g. page 10)
Conclusion paragraph is missing. I suggest to add this paragraph to highlight the limits of the revised articles and future directions in application of PPG technology for sleep analysis.
Author Response
Dear reviewer,
Thank you for your excellent comments that have helped us to improve the paper. We have carefully adapted and reworked the manuscript accordingly.
M Bruyneel
Authors propose a review article on PGG applications for sleep analysis.
The article is well written. However, in my opinion the first part of the article, that is devoted to explain the PPG technique, is too much specific, and not suitable for a general reader. Please try to be more clear.
> According to your suggestion, we have simplified the chapter: "2. Technical aspects of photoplethysmography"
Moreover, the introduction describe the history of PPG, but references are missing. Please add references for all the works that you have mentioned (e.g. The first attempts at developing instruments).
-> It is true that a lot of historical references are missing. Thank you for making us aware of this issue. We have now added them to the manuscript.
Moreover, at the end of paragraph 1.1 you mentioned the use of new techniques for PPG acquisition. Please add recent references (e.g. https://doi.org/10.1109/ISCC50000.2020.9219718)
-> This interesting work has been added to the manuscript, in a new chapter:
1.3. Recent developments in Photoplethysmography technology
Figures 1,2 and 3 are not clear. Please use good quality images. Furthermore, if these images have been already published in other works, please add a reference to them
-> These figures have not been copied from other works. According to the suggestion of another reviewer, we have merged the 3 figures.
The meaning of table 2 is not clear. Please add a description of this table content in the text.
-> Indeed, it does not add very much to the manuscript. We have removed this Table.
Please describe the process that you have followed to select the articles to include in this review.
-> This information has been added: "Now let’s review the recent academic studies focused on PPG signals-based algorithms to document sleep architecture in normal sleep. We have selected the studies that have compared data extracted from PPG with PSG in healthy adults, with the purpose of sleep/wake detection and/or classification of sleep stages."
Please proofread the article for typos and use the same font style (e.g. page 10)
-> Typos and inconsistent font style issues have been resolved.
Conclusion paragraph is missing. I suggest to add this paragraph to highlight the limits of the revised articles and future directions in application of PPG technology for sleep analysis.
-> Thank you for this excellent suggestion. We have added a conclusion paragraph.
Reviewer 3 Report
What was the key motivation behind focusing on the Photoplethysmography?
Authors should further clarify and elaborate novelty in their abstract.
Conclusion is too short. Add more explanation.
What are the limitations of the present work?
Authors should further clarify and elaborate novelty in their contribution.
What are the implications of this research?
Below papers has some interesting implications that you could discuss in your introduction and how it relates to your work.
Ijaz, Muhammad Fazal, Muhammad Attique, and Youngdoo Son. "Data-Driven Cervical Cancer Prediction Model with Outlier Detection and Over-Sampling Methods." Sensors 20.10 (2020): 2809.
Author Response
Dear reviewer,
Thank you for your excellent comments that have helped us to improve the paper. We have carefully adapted and reworked the manuscript accordingly.
M Bruyneel
What was the key motivation behind focusing on the Photoplethysmography?
-> The key motivation is now explained in the first paragraph of the Introduction section.
Authors should further clarify and elaborate novelty in their abstract.
-> Thank you for this excellent suggestion. We have reworked the abstract.
Conclusion is too short. Add more explanation.
-> It is true that Conclusion was really not appropriate. We have added more to the conclusion paragraph.
What are the limitations of the present work?
-> The fundamental limitations are now presented in the Discussion section.
Authors should further clarify and elaborate novelty in their contribution.
-> We have tried to clarify the fundamental interest in the topic of the paper and to emphasize the novelty in the Abstract and Discussion sections.
What are the implications of this research?
-> This point has also been developed in the Discussion section.
Below papers has some interesting implications that you could discuss in your introduction and how it relates to your work.
Ijaz, Muhammad Fazal, Muhammad Attique, and Youngdoo Son. "Data-Driven Cervical Cancer Prediction Model with Outlier Detection and Over-Sampling Methods." Sensors 20.10 (2020): 2809.
-> Thank you for suggesting this reading. We have added this reference to the text in the Introduction and Discussion sections.
Round 2
Reviewer 1 Report
Thanks to the authors for their improved manuscript.
I appreciate adding table 2 with examples of commercial products. However, since this is a research article, I would suggest to add a few cutting edge examples of such devices in terms of research and development.
I am not sure how reference 53 is relevant to this work. There are other examples from same group about PPG or wearable devices. See, https://doi.org/10.1002/admt.202000347
The quality of figure 1 should be improved. Texts are difficult to read.
Author Response
I appreciate adding table 2 with examples of commercial products. However, since this is a research article, I would suggest to add a few cutting edge examples of such devices in terms of research and development.
- We have added a section “Devices under development” to the Table 2
I am not sure how reference 53 is relevant to this work. There are other examples from same group about PPG or wearable devices. See, https://doi.org/10.1002/admt.202000347
- Thank you for providing this more adequate reference. We have changed this in the manuscript
The quality of figure 1 should be improved. Texts are difficult to read.
- We have reworked the Figure accordingly. I hope that the readability is better.
Reviewer 2 Report
The article has been strongly improved according with the reviewers' comments.
Author Response
Thank you very much
Reviewer 3 Report
.
Author Response
Thank you to have accepted our changes